# MSR-ViR: Modularized Self-reflected Video Reasoner for Video Question Answering

## Abstract

Recently, multimodal large language models (multimodal LLMs) have been applied to a wide range of video understanding tasks, particularly for Video Question Answering (VideoQA). However, existing multimodal LLMs suffer from the following challenge: the classic end-to-end training strategies of multimodal LLMs for VideoQA tasks are black-box, thus lacking interpretability as they can neither present a reasoning path nor indicate where the answer is derived from the video. To tackle this challenge, we propose **MSR-ViR** (**M**odularized **S**elf-**R**eflected **Vi**deo **R**easoner), a self-reflected framework that introduces a Modularized Spatial-Temporal Grounding (MoST-Grounding) module to multimodal LLMs for VideoQA tasks. MoST-Grounding utilizes a question parser LLM to generate execution policies, which serve as a reasoning path from questions to answers providing interpretability for our VideoQA framework. Based on the execution policies, MoST-Grounding invokes various small modules to localize temporal segments and spatial regions in videos which provide multimodal LLMs with most relevant visual information, while presenting visual evidence of our final answers. To avoid the question parser LLM generating unreasonable policies, we further propose a reinforcement learning-based Alternate Self-reflection training strategy to optimize the Multimodal LLM and the question parser LLM. Experiments on VideoQA datasets (NExT-QA and STAR) and grounded VideoQA dataset (NExT-GQA) demonstrate that our method significantly improves video understanding capabilities of multimodal LLMs, while providing interpretable reasoning paths together with temporal and spatial localization evidence within the video.

## 1 Introduction

Video Question Answering (VideoQA) is a representative task in video understanding, aiming to answer questions based on the content of a given video. Leveraging their rich external knowledge and strong generalization capabilities, multimodal large language models (multimodal LLMs) have emerged as powerful tools for tackling video understanding tasks such as VideoQA, video captioning and so on. Most multimodal LLMs encode frames from videos with visual encoders and utilize adapters to align the visual information with the textual query, allowing models to understand information in the video. However, these models face the following challenge in VideoQA tasks: the conventional end-to-end training approaches for multimodal LLMs operate as black-box systems, which inherently suffer from a lack of interpretability. Falling short in terms of transparency, they are unable to unveil the reasoning process or pinpoint the specific segments of the video from which the answers are derived.

To tackle this challenge, we propose the **M**odularized **S**elf-**R**eflected **Vi**deo **R**easoner (MSR-ViR) framework, including a **Mo**dularized **S**patial-**T**emporal **Grounding** (**MoST-Grounding**) module, together with a reinforcement learning-based **Alternate Self-reflection Training strategy** to train a multimodal LLM on VideoQA datasets, as shown in Figure 1(c). MoST-Grounding module first localizes the most relevant temporal segments and spatial regions in a video by utilizing various small modules according to the execution policy, which is generated by a question parser LLM. Then the spatial-temporal grounding results are encoded by a visual encoder and aligned with the textual inputs through a visual adapter, after which a Multimodal LLM utilizes the aligned information to

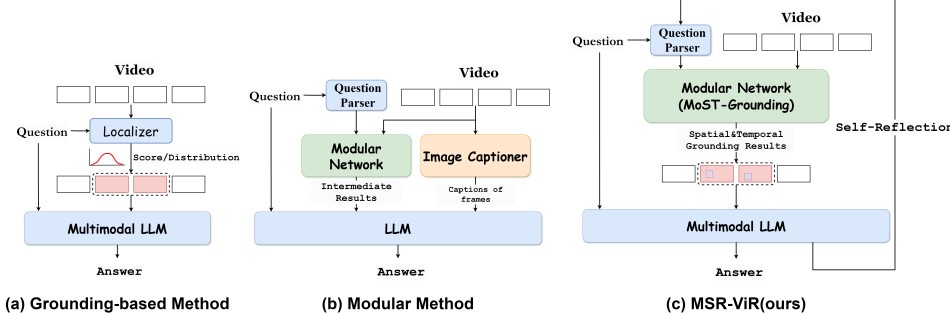

Figure 1: Comparison between existing works and our method.

generate the answer. The execution policy provides a clear reasoning path from the question to the answer, providing interpretability for our framework, while the spatial-temporal grounding results presenting visual evidence for our answers. Considering the question parser LLM that has not been supervised trained on relevant datasets may produce unreasonable or even incorrect policies, we further propose the Alternate Self-reflection Training Strategy. On one hand, the multimodal LLM undergoes Supervised Finetuning (SFT) based on the execution results of the policies generated by the question parser LLM to better understand the video content; on the other hand, the predicted result of the multimodal LLM also serves as a feedback to train the question parser LLM, where we adopt reinforcement learning due to the non-differentiable process when constructing the MoST-Grounding module. The training of the two LLMs alternates in a cyclical manner, and both LLMs are optimized during this alternate self-reflected training process.

We trained our multimodal LLM with both classic methods and our proposed approach on the training sets of commonly used VideoQA datasets, NExT-QA(Xiao et al., 2021) and STAR(Wu et al., 2021), and evaluated them on their corresponding test set. The results show that our approach outperforms classic training methods as well as other grounding-based VideoQA methods. We also conducted evaluations on NExT-GQA(Xiao et al., 2024), a widely-used grounding-based VideoQA dataset, and the results demonstrate that our method not only improves the performance of VideoQA but also more accurately localizes the temporal segments relevant to the questions compared to existing grounding-based methods and modular methods.

Our contributions can be summarized as follows:

- We propose **MSR-ViR**, a self-reflected VideoQA framework with multimodal LLMs which utilizes a **MoST-Grounding** module to locate the most relevant spatial-temporal information in the video, together with parser LLM to generate modular interpretable execution policies.

- We propose a reinforcement learning-based **Alternate Self-reflection Training Strategy** to train the multi-modal LLM and parser LLM jointly.

- We conducted experiments on commonly used VideoQA datasets to demonstrate that our method outperforms classic training methods and other grounding-based methods. Additionally, experiments on the grounded VideoQA dataset show that our approach could localize the temporal segments accurately, providing visually-grounded evidence of the final answer.

## 2 RELATED WORKS

**Video Understanding with multimodal LLMs.** With the development of multimodal LLMs, they have been utilized for video understanding tasks(Zhang et al., 2023b; Lin et al., 2023a; Maaz et al., 2023; Li et al., 2023b; 2024b; Zhang et al., 2024; Song et al., 2024; Yao et al., 2024; Li et al., 2024a). Most multimodal LLMs for videos are built on open-source LLMs such as LLaMA(Touvron et al., 2023) and Vicuna(Chiang et al., 2023), and adapters are utilized to align encoded visual information with the textual space. Representative works like Video-LLaMA(Zhang et al., 2023b)

utilizes the Vision Transformer (ViT) from EVA-CLIP(Sun et al., 2023) and an Image Q-Former(Li et al., 2023a) as the video frame encoder, after which a Video Q-Former is employed to encode temporal information in the video, with a linear layer projecting this visual information into the textual space. Some other works have explored the temporal perception capabilities of multimodal LLMs(Huang et al., 2024; Ren et al., 2024; Wang et al., 2024c; Qian et al.; Li et al., 2024c) to enhance their understanding of temporal information in videos. For instance, VTimeLLM(Huang et al., 2024) introduces a boundary perception training process on multi-event datasets, improving the performance of multimodal LLMs on tasks such as video grounding and dense video captioning. However, classic end-to-end training methods of multimodal LLMs remain black boxes, resulting in a lack of interpretability as they are unable to provide inference process as well as grounded evidence of the answer in the video.

**Grounded VideoQA with LLMs.** Grounded VideoQA aims to answer the question and at the same time indicates where in the video the answer originates. Existing grounding-based(retrieval-based) VideoQA methods(Wang et al., 2024d; Xiao et al., 2024; Qian et al., 2024; Yu et al., 2024; Wang et al., 2024a) attempt to localize time segments relevant to the question within the video in the first place and then sample frames from the identified segments to serve as inputs to multimodal LLMs, as is shown in Figure 1(a). For example, LSTP(Wang et al., 2024d) leverages optical flow of videos to efficiently extract relevant video frames as visual input of multimodal LLM to achieve long-form video-text understanding. Grounded-based methods address the issue of providing visual evidence in videos to some extent, but still they lack interpretability as they typically rely on black-box models to perform temporal localization without a clear reasoning path, especially for questions with complicated structures.

**Modular VideoQA with LLMs.** Modular methods(Min et al., 2024; Zhang et al., 2023a; Surís et al., 2023; Wang et al., 2024b;e) utilize various smaller models according to execution policies generated by certain LLM to handle sub-tasks derived from the original complex question, and another LLM integrates the outputs of these smaller models to produce the final answer, as shown in Figure 1(b). MoReVQA(Min et al., 2024) utilizes multiple PaLM-2(Anil et al., 2023) LLMs in a multi-stage modular reasoning process, where the reasoning results and video frame captions obtained from an image captioner are provided to another PaLM-2 model to derive the final answer. ViperGPT(Surís et al., 2023) uses a code-finetuned LLM as a python program generator to generate programs that invokes various tools and apis provided in prompts. While this approach enhances interpretability, the unimodal LLMs used can only receive video information in the form of captions generated from original video frames, potentially overlooking details in individual frames and missing important temporal context between frames. Additionally, the lack of self-reflection may cause the modular network to execute according to unreasonable policies generated by LLMs, thereby affecting the accuracy of question answering.

## 3 METHOD

In this section, we describe our proposed framework MSR-ViR(Modularized Self- Reflected Video Reasoner), a modularized videoQA framework with alternate self-reflection training strategy. Figure 2 demonstrates the overall framework, together with an example for illustration. We will first introduce the question parser which generates modularization policies in Section 3.1, and then introduce our MoST-Grounding module designed for temporally and spatially grounding visual clues according to the generated policy in Section 3.2. In Section 3.3, we will present how to processes various information from both MoST-Grounding module and naive inputs(the video and the question) based on a multimodal LLM. Finally, our proposed Alternate Self-reflection Training Strategy is introduced in Section 3.4.

### 3.1 QUESTION PARSER

Many video language questions actually involve a "multi-step" reasoning process rather than the end-to-end "one-step" processing. As the example shown in Figure 2, when asked "Why does the man have to squat down after the car approaches?", we first need to locate the relevant temporal segment "after the car approaches". Next, we must identify the man who is "squatting down", and finally determine the reason to answer the question. Similarly, our MSR-ViR framework mirrors the above "multi-step" reasoning process how humans tackle VideoQA tasks: when facing a video

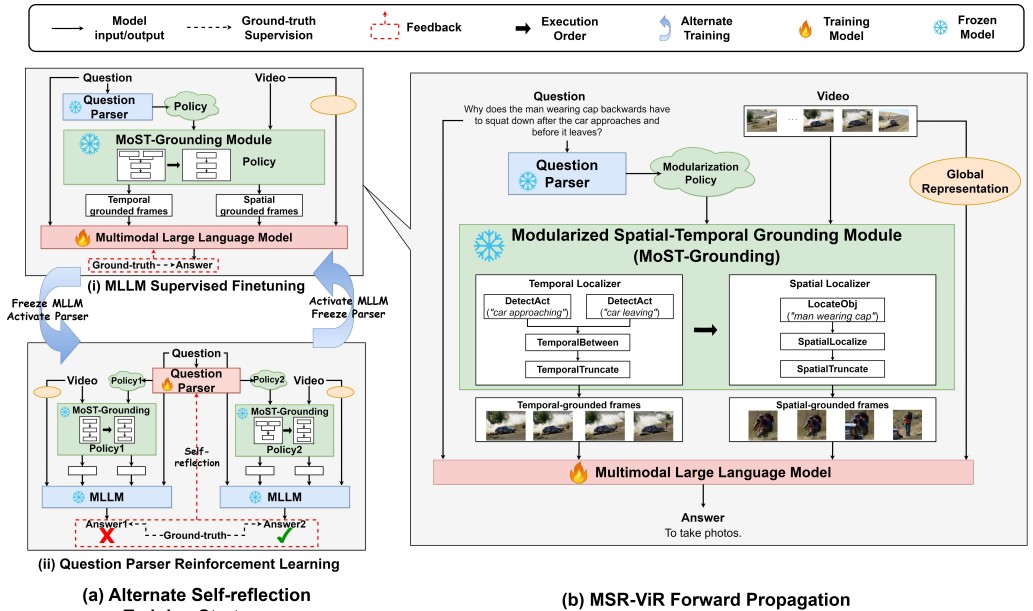

Figure 2: Framework of MSR-ViR. The left part (a) shows Alternate Self-reflection Training Strategy, including (i) Multimodal LLM Supervised Finetuning and (ii) Question Parser Reinforcement Learning. The right part (b) demonstrates forward propagation details of MSR-ViR during MLLM supervised fine-tuning.

together with a complex question, we first utilize a **Question Parser** to decompose the question into several sub-questions, allowing us to identify the relevant video segments and regions, together with a tree-structured reasoning process, to explicitly help answer the question.

Given a question $q$, our question parser $Q$ aims to generate the policy $p = Q(q)$, which serves as the execution plan for the subsequent MoST-Grounding module (illustrated in Sec. 3.2) to invoke various small modules for temporal and spatial localization. Considering the diversity of question content and structure, as well as the limited training data for question policies, we utilize a large language model as the question parser and take advantage of its in-context learning capabilities (Brown et al., 2020). As shown in Figure 3, we carefully design the prompt and stimulate question parser to generate policies in the uniform JSON format, organizing and chaining the small modules in a specific structure. The JSON structure allows MoST-Grounding module to recursively call each module, ultimately generating the spatial-temporal grounding results. The complete prompt for question parser is presented in appendix A.1.

## 3.2 MODULARIZED SPATIAL-TEMPORAL GROUNDING MODULE

MoST-Grounding module is the core component of our framework, recursively invoking various small modules according to the modular policy generated by the question parser to achieve temporal and spatial localization for complex questions. Our MoST-Grounding module consists of two parts: **temporal localizer** $\mathcal{F}_t$ and **spatial localizer** $\mathcal{F}_s$, each containing several small modules for temporal and spatial localization, respectively.

Given a concept $c_t$ ("car approaching", for example) and a video $v = \{v_1, v_2, \ldots, v_T\}$ containing $T$ frames, temporal localizer aims to generate the most relevant segments $v_s = \{v_i, \ldots, v_j\}$ from the video $v$, where $1 \leq i \leq j \leq T$. Later the output of the temporal localizer, namely video segments $v_s$, along with concepts $c_s$ are processed to the spatial localizer, generating most relevant visual bounding box $b_{v_s}$ within video segments $v_s$. Formally, MoST-Grounding module $\mathcal{M}$ is written as follows:

$$\mathcal{M}(v, c_t, c_s) = \mathcal{F}_s\big(\mathcal{F}_t(v, c_t), c_s\big),$$  (1)

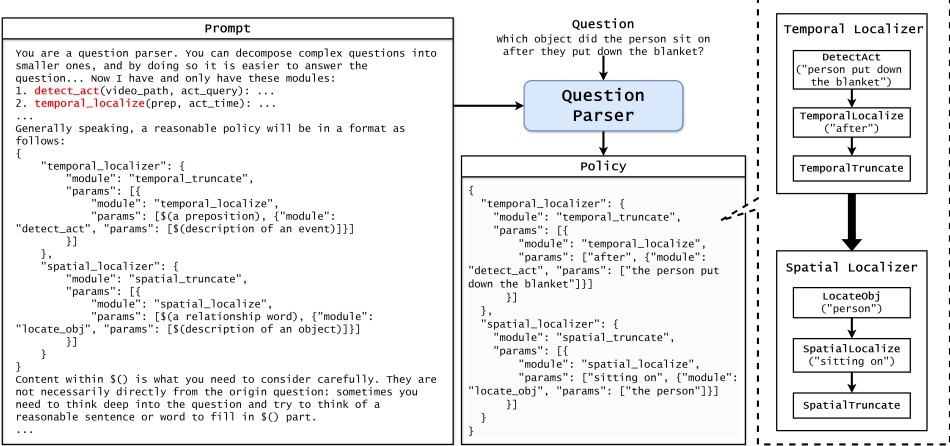

Figure 3: Prompt for question parser LLM with an example policy.

To address complex semantic scenarios, both the temporal localizer and spatial localizer consist of several types of small modules that can be dynamically assembled according to the policy. To be specific, there are 7 types of small modules in our MoST-Grounding module, 4 of which belong to temporal localizer and the other 3 belong to spatial localizer. Temporal localizer deals with temporal information provided in the question. As the core module of temporal localizer, `DetectAct` module temporally localizes a simple action described by a short query("car approaching", for example) in the video. In this module we utilize UniVTG(Lin et al., 2023b), a unified video temporal grounding framework with zero-shot grounding ability over diverse action queries and fast inference speed. Spatial localizer on the other hand perceives spatial relationships within the video, with a `LocateObj` module localizing an object described by a short query("man wearing cap", for example) in a video frame. In this module we take advantages of YOLO-World(Cheng et al., 2024), an open-vocabulary object detection framework that achieves real-time detection. Details about other modules are presented in appendix A.2.

In our execution policies, both temporal localizer and spatial localizer will dynamically assemble corresponding modules $\{m_i^t\}$ and $\{m_i^s\}$, respectively. With the policy $p$ generated by question parser, $\mathcal{F}_t$ and $\mathcal{F}_s$ in Eq. 1 would be instantiated as follows:

$$\mathcal{F}_{t|p}(\cdot) = \mathcal{I}\big(\{m_i^t\}, p\big)(\cdot), \mathcal{F}_{s|p}(\cdot) = \mathcal{I}\big(\{m_i^s\}, p\big)(\cdot), \tag{2}$$

After instantiating with the policy, the modules within the temporal localizer are called first to locate and extract the temporal-grounded frames from the video. Subsequently, the modules within the spatial localizer are invoked to generate the corresponding spatial-grounded frame for each temporal-grounded frame. In this modular manner, MoST-Grounding extracts several temporally and spatially localized video frames from the video, which will serve as visual input to our multimodal LLM.

### 3.3 MULTIMODAL LLM ANSWERER

After MoST-Grounding localizes the temporal segments and spatial regions relevant to the question, a multimodal LLM is needed to understand the textual and visual information in order to answer the question. Formally, the answer of a question $q$ given the video $v$ can be written as follows:

$$\hat{y}(q, v) = \mathcal{F}\big(q, v_s, b_{v_s}\big), \tag{3}$$

where $v_s$ and $b_{v_s}$ represent the video segments and bounding box generated from MoST-Grounding module, and $\mathcal{F}(\dots)$ denotes the forward propagation of multimodal LLM. To better enhance the video understanding ability of the multimodal LLM, we extend the input of the multimodal LLM in Eq. 3 with the following two strategies. Firstly, we provide an additional global representation of the video to the multimodal LLM. This is necessary because the MoST-Grounding module may not always accurately localize the segments relevant to the question, and the multimodal LLM might overlook essential information if it relies solely on the grounding results. Specifically, we sample several frames from the video at regular intervals and perform average pooling on the encoded video

frames along the temporal dimension to create a compressed representation of the global video, which is then input alongside the grounding results. Secondly, in addition to the original question and the aforementioned visual information, we provide the multimodal LLM with a guiding prompt that explains the specific meanings of various visual input components. With these two designs, Eq. 3 would be modified with:

$$\hat{y}(q, v) = \mathcal{F}\big(Pt(q, v_s, b_{v_s}, g_v)\big), \tag{4}$$

where $Pt(\dots)$ represents the guiding prompt and $g_v$ denotes the global video representation. Specific input format of the multimodal LLM is presented in appendix A.3.

With the ground-truth of question $q$ being $y$, the supervised finetuning loss of the multimodal LLM is defined as a cross entropy loss $\mathcal{L}_{\text{CE}}$:

$$\mathcal{L}_{\text{CE}}(\hat{y}(q, v), y) = - \sum_{(v,q,y) \in D} y \log(\hat{y}(q, v)) \tag{5}$$

where $y$ is the target answer, $D$ is the dataset. By optimizing the loss function in Eq. 5, the multimodal LLM undergoes supervised finetuning on VideoQA datasets, learning to answer questions based on all the provided information.

## 3.4 ALTERNATE SELF-REFLECTION TRAINING STRATEGY

As discussed in Section 3.1, we can teach the question parser to generate modularization policies from complex questions by providing examples in the prompt. However, relying solely on in-context learning does not ensure the quality of the policies. To address this issue, we propose the Alternate Self-Reflection Training Strategy, which enables the question parser to improve the quality of its policies through reinforcement learning.

We assume that for a given question, provided that all our modules remain unchanged, a reasonable modular policy is more likely to accurately localize the correct temporal segments and spatial regions. Consequently, the loss computed by the multimodal LLM is likely to be smaller. In contrast, an unreasonable policy that leads to incorrect localization will result in an increase in this loss. Therefore, we provide feedback to the question parser LLM using the loss noticed during the training process of the multimodal LLM, thereby guiding it through reinforcement learning training. We utilize Direct Preference Optimization(DPO)(Rafailov et al., 2024), a method for Reinforcement Learning from Human Feedback (RLHF), to train the question parser LLM to generate more reasonable policies. Different from previous RLHF methods, DPO directly optimizes a language model without explicit rewarding models, making the training process straight-forward and stable. Specifically in DPO, the policy objective is formulated as:

$$\mathcal{L}_{\text{DPO}}(\pi_\theta; \pi_{\text{ref}}) = -\mathbb{E}_{(q, p_w, p_l) \sim \mathcal{D}} \left[ \log \sigma \left( \beta \log \frac{\pi_\theta(p_w \mid q)}{\pi_{\text{ref}}(p_w \mid q)} - \beta \log \frac{\pi_\theta(p_l \mid q)}{\pi_{\text{ref}}(p_l \mid q)} \right) \right] \tag{6}$$

where in our case, $p_w$ denotes the positive policy, $p_l$ denotes the negative policy and $q$ is the input question. $\pi_\theta$ is our question parser LLM to be trained, while $\pi_{\text{ref}}$ is a reference model initialized with our question parser LLM but remain frozen. $\sigma$ is the sigmoid function, and $\beta$ is a controlling parameter. Through DPO training, the probability of generating positive policies increases, while the probability of generating negative policies decreases. In other words, the question parser LLM learns to generate more reasonable policies.

We prompt the question parser LLM to view the same question from multiple perspectives, generating different modular policies. The MoST-Grounding module executes each policy, producing their respective grounding results. The multimodal LLM then computes the corresponding losses. We classify the policy with the smaller loss as positive and the one with the larger loss as negative, training the question parser LLM according to Eq. 6.

Our training strategy alternates between SFT of the multimodal LLM and reinforcement learning for the question parser LLM, optimizing with the loss functions in Eq. 5 and Eq. 6, respectively. While training one large model, the other model's parameters remain frozen. During this process, the multimodal LLM periodically pauses to adapt based on the modular policies from the question parser LLM. After a set training period, the question parser LLM utilizes these refined policies to further train the multimodal LLM, allowing both models to optimize continuously. See appendix A.4 for detailed training process.

## 4 EXPERIMENTS

In this section, we introduce our experiments in the following four parts: we first introduce the basic setups of our experiments, including datasets, baselines and our implementation details. Next, we introduce our experiments on NExT-QA(Xiao et al., 2021) and STAR(Wu et al., 2021) datasets to show that our MSR-ViR framework enhances the performance of Multimodal LLM on VideoQA tasks. Then, we present our experiments on NExT-GQA(Xiao et al., 2024) dataset to demonstrate that MSR-ViR grounds the temporal segments relevant to the questions more accurately than existing grounding-based methods and modular methods. Finally, we present the ablation study and qualitative analysis.

### 4.1 EXPERIMENTS SETUPS

**Datasets.** We conducted our experiments on two VideoQA datasets: NExT-QA(Xiao et al., 2021) and STAR(Wu et al., 2021), together with a grounded VideoQA datasets NExT-GQA(Xiao et al., 2024). NExT-QA is a multi-choice VideoQA dataset focused on temporal actions including three types of questions: **Temporal**, **Causal** and **Descriptive**. STAR is a multi-choice VideoQA dataset for situated reasoning in real-world videos that contains four types of questions: **Interaction**, **Sequence**, **Prediction** and **Feasibility**. For STAR, we created a subset (STAR-sub) with **Interaction** and **Sequence** questions (82.5% of STAR), excluding **Prediction** and **Feasibility** types as they lack temporal and spatial grounding in videos, making them unsuitable for our framework. NExT-GQA is derived from NExT-QA, retaining **Temporal** and **Causal** questions but removes **Descriptive** ones, providing ground-truth temporal clips for validation and test sets to evaluate temporal grounding accuracy.

**Baselines.** On NExT-QA and STAR, our baselines include vision-language models AIO(Wang et al., 2023), ATP(Buch et al., 2022), VGT(Xiao et al., 2022), MIST(Gao et al., 2023), VFC(Momeni et al., 2023), CoVGT(Xiao et al., 2023), HiTeA(Ye et al., 2023), InternVideo(Wang et al., 2022), multi-modal LLMs BLIP2(Li et al., 2023a), InstructBLIP(Dai et al., 2023) and grounding-based multi-modal LLMs LSTP(Wang et al., 2024d), SeViLa(Yu et al., 2024), GCG(Wang et al., 2024a). As the direct baseline of our framework, we train Qwen-VL(Bai et al., 2023) and Llava-Next(Zhang et al., 2024) by uniformly sampling frames from videos. On NExT-GQA dataset, our baselines include vision-language models VGT, VIOLETv2(Fu et al., 2023), Temp[CLIP](Radford et al., 2021), FrozenBiLM(Yang et al., 2022)(which achieve grounded VideoQA with the method in Xiao et al. (2024)), grounding-based method LSTP, SeViLa, LangRepo(Kahatapitiya et al., 2024) and modular method LLoVi(Zhang et al., 2023a), MoReVQA(Min et al., 2024). We present zero-shot results for LangRepo, LLoVi and MoReVQA, while conducting weakly-supervised finetuning on NExT-GQA training set for other baselines.

**Implementations.** We implement our method based on SWIFT framework(Zhao et al., 2024). We utilize a large language model, Qwen2-7B(Yang et al., 2024) as the backbone for our question parser, and we utilize Qwen-VL(Bai et al., 2023) and Llava-Next(Zhang et al., 2024) as the backbone for our multimodal LLM, denoted as MSR-ViR$_Q$ and MSR-ViR$_L$ respectively. Following the classic training strategy, we uniformly sample 4 frames from videos for Qwen-VL and 8 frames for Llava-Next to implement our direct baseline in Table 1. For MSR-ViR$_Q$, we sample 2 frames from temporal grounding results, and 2 corresponding spatial-grounded frames, while for MSR-ViR$_L$ we sample 8 frames from temporal grounding results, 8 spatial-grounded frames accordingly. We utilize LoRA(Hu et al.) during supervised finetuning of our Multimodal LLM. As for our Alternate Self-reflection Training Strategy, the period for alternating training between two LLMs is 200 steps, with the gradient accumulation step set to 16. We conduct 5 epochs of SFT on every training set for Qwen-VL, Llava-Next and our MSR-ViR framework, selecting the best model according to the results on validation set.

### 4.2 EXPERIMENTS ON VIDEOQA

We compare our MSR-ViR framework with existing vision-language models, multimodal LLMs and grounding-based methods on NExT-QA and STAR-sub. As shown in Table 1, MSR-ViR$_L$ achieves best results on the overall NExT-QA and STAR-sub dataset together with most subsets formed by different types of questions. Particularly, MSR-ViR$_L$ surpasses LSTP and SeviLa, which also utilize

Table 1: Experiments on **NExT-QA** and **STAR-sub**. All models are finetuned on the corresponding training set.The first part contains small vision-language models, and in the second part models or methods are based on multimodal LLMs. Qwen-VL and Llava-Next are our direct baselines, MSR-ViR$_Q$ is our framework based on Qwen-VL and MSR-ViR$_L$ is our framework based on Llava-Next. **Bold** number denotes the best result.

| Method | NExT-QA | | | | STAR-sub | | |
|---|---|---|---|---|---|---|---|
| | Temporal | Causal | Descriptive | Avg. | Interaction | Sequence | Avg. |
| AIO | 48.6 | 48.0 | 63.2 | 50.6 | 47.5 | 50.8 | 49.2 |
| ATP | 49.3 | 48.6 | 65.0 | 51.5 | 50.6 | 52.8 | 51.7 |
| VGT | 55.0 | 52.2 | 64.0 | 55.0 | - | - | - |
| MIST | 56.6 | 54.6 | 66.9 | 57.1 | 55.5 | 54.2 | 54.9 |
| VFC | 53.3 | 57.6 | 72.8 | 58.6 | - | - | - |
| CoVGT | 57.4 | 58.8 | 69.3 | 60.0 | - | - | - |
| HiTeA | 58.3 | 62.4 | 75.6 | 63.1 | - | - | - |
| InternVideo | 58.5 | 62.5 | 75.8 | 63.2 | 62.7 | 65.6 | 64.4 |
| BLIP-2 | 64.9 | 69.7 | 79.4 | 69.6 | - | - | - |
| LSTP | 66.5 | 72.8 | 81.2 | 72.1 | - | - | - |
| InstructBLIP | 70.5 | 71.5 | 79.8 | 72.5 | - | - | - |
| SeViLa | 69.4 | 74.2 | 81.3 | 73.8 | 63.7 | 70.4 | 67.1 |
| GCG | **72.6** | 74.2 | 80.7 | 74.6 | - | - | - |
| Qwen-VL | 68.4 | 71.3 | 80.6 | 71.9 | 60.4 | 65.5 | 63.0 |
| Llava-Next | 69.5 | 73.3 | 79.7 | 73.1 | 67.6 | 72.1 | 69.9 |
| MSR-ViR$_Q$(ours) | 69.9 | 73.4 | **81.5** | 73.6 | 64.8 | 68.0 | 66.4 |
| MSR-ViR$_L$(ours) | 72.2 | **74.6** | 80.9 | **74.9** | **68.9** | **73.1** | **71.0** |

grounding-based multimodal LLMs, on **Temporal** questions where temporal information is essential, demonstrating the superior temporal understanding ability of our method comparing to previous grounding-based methods. Besides, for **Interaction** questions where spatial information is relatively important, our method MSR-ViR$_L$ also presents the best performance. Comparing MSR-ViR$_Q$ and MSR-ViR$_L$ with their own direct baseline that has been trained with uniformly sampled frames in videos, we prove that our framework help enhance VideoQA abilities of multimodal LLMs by providing them with most relevant grounded information, ignoring redundant information that may impair understanding.

## 4.3 EXPERIMENTS ON GROUNDED VIDEOQA

To further confirm MSR-ViR is capable of more accurately grounding the relevant information thus enhancing VideoQA ability of multimodal LLMs, we conducted experiments on the NExT-GQA(Xiao et al., 2024) dataset. NExT-GQA not only contains the answer to the question, but also presents a human-annotated ground-truth time span, indicating where the answer is derived from the video, in other words, the most relevant time period to the question. The dataset requires VideoQA models to provide "evidence" of their answer, evaluating the grounding accuracy with **IoP**(Intersection over Prediction) and **IoU**(Intersection over Union). It also measures **Acc@GQA**, which is the proportion of questions that are correctly answered, and at the same time **IoP** between predicted time span and ground-truth time span is larger than 0.5.

We compare our MSR-ViR framework with existing grounding-based methods and modular methods, together with vision-language models and multimodal LLMs which utilizes **NG+** method in (Xiao et al., 2024) for training. The results are shown in Table 2. Methods in the first part are models implemented with **NG+**, while the second part includes grounding-based methods and the third part contains modular methods. We de-emphasize methods implemented with significantly larger LLMs(LLoVi with GPT-4 for example) for fair comparison. MSR-ViR$_Q$ achieves best results

Table 2: Experiments on **NExT-GQA**. VGT, VIOLETv2, Temp[CLIP], FrozenBiLM, LSTP, SeViLa, MSR-ViR$_Q$(ours) and MSR-ViR$_L$(ours) are finetuned on the training set. Acc@GQA is the grounded QA accuracy defined in Xiao et al. (2024). Method in gray lines utilize significantly larger LLMs(Palm-2 and GPT-4). **Bold** number denotes the best result excluding gray methods.

| Method | mIoP | IoP @0.3 | IoP @0.5 | mIoU | IoU @0.3 | IoU @0.5 | Acc@GQA |
|---|---|---|---|---|---|---|---|
| VGT | 25.3 | 26.4 | 25.3 | 3.0 | 3.6 | 1.7 | 14.4 |
| VIOLETv2 | 23.6 | 25.1 | 23.3 | 3.1 | 4.3 | 1.3 | 12.8 |
| Temp[CLIP] | 25.7 | 31.4 | **25.5** | 12.1 | 17.5 | 8.9 | 16.0 |
| FrozenBiLM | 24.2 | 28.5 | 23.7 | 9.6 | 13.5 | 6.1 | 17.5 |
| LSTP | - | - | - | 19.9 | 23.3 | 11.2 | - |
| LangRepo | 20.3 | - | 20.0 | 8.7 | - | 6.0 | 11.2 |
| SeViLa | 29.5 | 34.7 | 22.9 | 21.7 | 29.2 | 13.8 | 16.6 |
| LLoVi(Mistral-7B) | 20.7 | - | 20.5 | 8.7 | - | 6.0 | 11.2 |
| LLoVi(GPT-4) | 37.3 | - | 36.9 | 20.0 | - | 15.3 | 24.3 |
| MoReVQA(Palm-2) | 37.8 | - | 37.6 | 19.7 | - | 15.4 | 39.6 |
| MSR-ViR$_Q$(ours) | **30.0** | **39.8** | 25.0 | 22.8 | 33.0 | **16.4** | 18.5 |
| MSR-ViR$_L$(ours) | 29.6 | 39.0 | 24.1 | **23.4** | **33.6** | **16.4** | **18.6** |

on mIoP, while MSR-ViR$_L$ achieves best results on mIoU, indicating that our method grounds the temporal segment relevant to the question more precisely than existing grounding-based VideoQA methods and modular VideoQA methods. The best result of Acc@GQA demonstrates that our method can perform VideoQA tasks more effectively, while also providing more reasonable temporal evidence indicating which specific segment of the video the answer derives from.

## 4.4 ABLATION STUDY

As demonstrated in Table 3, to further validate the effectiveness of modules and designs in our MSR-ViR framework, we conduct ablation study on NExT-QA and NExT-GQA dataset for MSR-ViR$_Q$ concerning the following questions:

**Is Alternate Self-reflection Training Strategy necessary?** We remove the self-reflection training process, only finetuning our Multimodal LLM without training the question parser LLM, and the results are shown by w/o self-reflection in Table 3. The average accuracy on NExT-QA declines by 1.5, and accuracy on each subsets decreases to varying degrees. The grounded accuracy as well as IoU of temporal grounding also decline as shown in experiments on NExT-GQA. This demonstrates the necessity of our Alternate Self-reflection Training Strategy.

**Is spatial_localizer necessary in MoST-Grounding module?** Most existing grounding-based methods only consider temporal grounding, so we remove the **spatial_localizer** including all small modules in it, only providing our Multimodal LLM with temporal grounding results, denoted by w/o spatial modules. The average accuracy on NExT-QA drops by 1.4, proving that spatial grounding results provided by **spatial_localizer** contain useful information for Multimodal LLM to answer the question correctly.

**Are our designs in training Multimodal LLM necessary?** In 3.3, we introduce two designs for our Multimodal LLM training: global representation and instruction prompts. We removed these two designs separately and conducted tests on NExT-QA. The results show that the average accuracy on NExT-QA has decreased to varying degrees for both, indicating that the two designs we proposed for training multimodal LLMs are effective.

Furthermore, we visualize the reasoning process of MSR-ViR through an example, as demonstrated in Figure 4. After self-reflection training, question parser generates a more reasonable policy accurately grounding the question in the video, leading to a correct answer. More examples can be found in appendix A.5.

Table 3: Ablation study on NExT-QA and NExT-GQA. **Bold** number denotes the best result.

| | NExT-QA | | | |
|---|---|---|---|---|
| | **Temporal** | **Causal** | **Descriptive** | **Average** |
| MSR-ViR$_Q$ | **69.9** | **73.4** | 81.5 | **73.6** |
| w/o self-reflection | 67.2 | 72.5 | 80.5 | 72.1 |
| w/o spatial modules | 67.0 | 72.5 | 81.4 | 72.2 |
| w/o instruction prompts | 68.3 | 72.4 | **82.4** | 72.8 |
| w/o global representation | 66.9 | 70.1 | 78.0 | 70.4 |
| | NExT-GQA | | | |
| | **Acc@QA** | **Acc@GQA** | **mIoU** | **IoU@0.5** |
| MSR-ViR$_Q$ | **69.9** | **18.5** | **22.8** | **16.4** |
| w/o self-reflection | 68.3 | 17.9 | 22.2 | 15.7 |

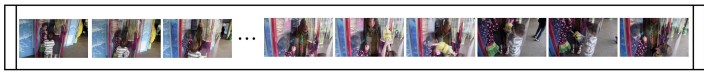

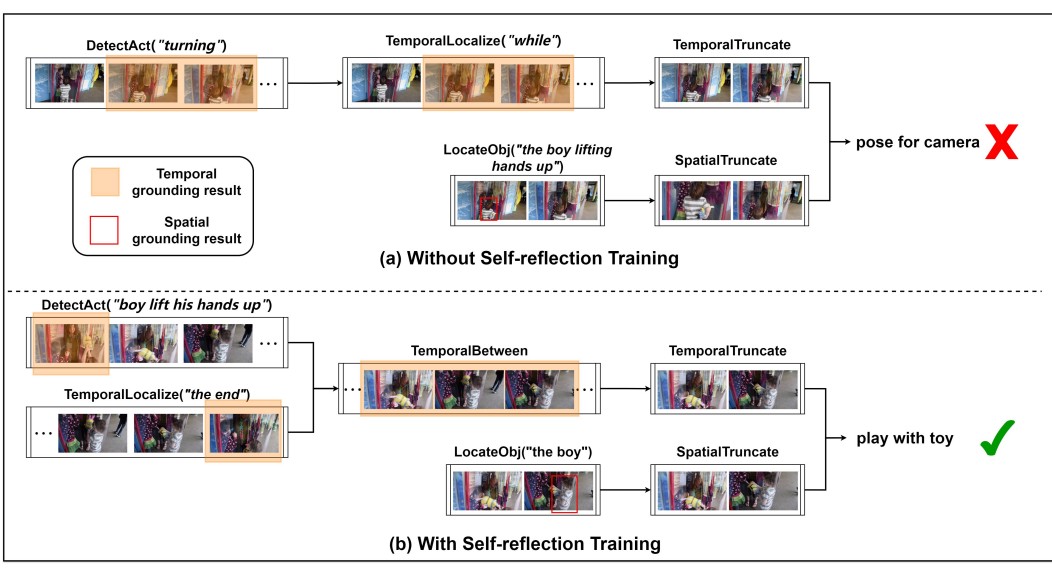

Figure 4: Visualization of MSR-ViR reasoning process. (a) is the inference process of MSR-ViR without self-reflection training. (b) is the inference process of MSR-ViR with self-reflection training.

## 5 CONCLUSION

In summary, we propose Modularized Self-Reflected Video Reasoner(MSR-ViR), a self-reflected framework that integrates a Modularized Spatial-Temporal Grounding(MoST-Grounding) module into a Multimodal LLM for interpretable VideoQA. Modularization policies generated by a question parser LLM demonstrates clear paths from questions to answers, enhancing interpretability of our framework, while spatial-temporal grounding results present visual evidence for answers. Through the proposed alternate self-refection training process, policies are gradually refined, becoming more reasonable. Extensive experiments demonstrate that MSR-ViR significantly improves VideoQA capabilities of multimodal LLMs while grounding answers in videos more accurately. Future work could explore further enhancements to the design of modular network and its execution efficiency.

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

## A APPENDIX

Table 4: Illustration of our modules. **Type** is the type of the module, where T represents modules in **temporal_localize**r and S represents modules in **spatial_localizer**. **Vision-Language Model** denotes the small vision-language model we use in the module. As for the **input** and **output**, $V$ denotes the input video, $I$ denotes the video frame, $TS$ denotes a time period, $BBOX$ denotes a bounding box, $Q_a$ denotes the action query, $Q_o$ denotes the object query, $p$ is a preposition representing certain temporal relationship and $r$ is a word or phrase representing certain spatial relationship.

| Module Name | Type | Vision-Language Model | Input | Ouput |
|---|---|---|---|---|
| DetectAct | T | UniVTG | $V, Q_a$ | $TS$ |
| TemporalLocalize | T | - | $p, TS^{(in)}$ | $TS^{(out)}$ |
| TemporalBetween | T | - | $TS_1, TS_2$ | $TS^{(out)}$ |
| TemporalTruncate | T | - | $V, TS$ | $I_1, I_2, ...$ |
| LocateObj | S | YOLO-World | $I, Q_o$ | $BBOX$ |
| SpatialLocalize | S | YOLO-World | $I, BBOX^{(in)}, r$ | $BBOX^{(out)}$ |
| SpatialTruncate | S | - | $I, BBOX_1, BBOX_2, ...$ | $I^{(out)}$ |

## A.1 PROMPT FOR QUESTION PARSER LLM

We carefully design a prompt for our question parser LLM to generate policies from given questions from in-context learning. The complete prompt is presented in Figure 5, 6, 7. We first inform the question parser of some basic information and an introduction to the functions of each module. Then we tell it the general template of the policy and several variants under special circumstances. Finally, through a few examples, we teach the question parser how to generate a policy based on the question.

```
You are a question parser. You can decompose complex questions into smaller ones, and by doing so it is easier to
answer the question. The question is given together with a video clip, and now I hope to temporally and spatially ground
the question in the video clip before answering it. The question parser need to decompose the question into several
modules, and by solving each module step by step, the question can be grounded in the video. Now I have and only have
these modules:
1. detect_act(video_path, act_query): This module works as a temporal localizer. Provided with the path of a video and a
query of an action, the module will temporally locate the query into the video, and return a timespan: [start_time,
end_time];
2. temporal_localize(prep, act_time): This module locate a timespan in the video. act_time is usually the return value of
detect_act, which looks like [start_time, end_time], and prep is a preposition ("before", "when", "after" or something like
that). The module determine the final timespan according to the preposition and the act_time, also returning in the format
of [start_time, end_time];
3. temporal_between(act_time1, act_time2): This module is actually a special form of temporal_localize, where the
preposition is "between". So act_time1 and act_time2 are two timespans detected by "detect_act" module, and this
module locate the timespan between the two actions, resulting in returning [start_time, end_time] as well.
4. temporal_truncate(video_path, tp_result): This module truncate a clip of video according to the provided timespan
tp_result: [start_time, end_time]. The module samples 2 frames between the start_time of the video and the end_time of
the video, thus returning a list of frames: [frame1, frame2];
5. locate_obj(image, obj_query): This module works as a spatial localizer, which spatially locate an area in the image
according to the provided obj_query, returning a bounding box of the area: [x1, y1, x2, y2].
6. spatial_localize(image, rel, sp_result): This module locate a specific area in an image. sp_result is the return value of
module "locate_obj" which is a bounding box, and rel is a verb or preposition that describes the relationship between the
expected result and the given sp_result. The return of this module is the expected result that is also a bounding box.
7. spatial_truncate(image, sp_result): This module truncate the bounding box sp_result in the provided image, resize the
bounding box area to the same size of the original image, and returning the resized image.
Among all these modules, 1, 2, 3, 4 are temporal modules and 5, 6, 7 are spatial modules. What you need to do is to
provide a policy according to the given question using the above 7 modules. A policy always has two parts:
temporal_localizer and spatial_localizer, each utilizing one or some of the above 7 modules.
Generally speaking, a reasonable policy will be in a format as follows:
{
    "temporal_localizer": {
        "module": "temporal_truncate",
        "params": [{
            "module": "temporal_localize",
            "params": [$(a preposition), {"module": "detect_act", "params": [$(description of an event)]}]
        }]
    },
    "spatial_localizer": {
        "module": "spatial_truncate",
        "params": [{
            "module": "spatial_localize",
            "params": [$(a relationship word), {"module": "locate_obj", "params": [$(description of an object)]}]
        }]
    }
}
Content within $() is what you need to consider carefully. They are not necessarily directly from the origin question:
sometimes you need to think deep into the question and try to think of a reasonable sentence or word to fill in $() part.
If you think there are no temporal event that needs to be specifically located, just simplify the format as:
{
    "temporal_localizer": {
        "module": "temporal_truncate",
        "params": []
    },
    "spatial_localizer": {
        "module": "spatial_truncate",
        "params": [{
            "module": "spatial_localize",
            "params": [$(a relationship word), {"module": "locate_obj", "params": [$(description of an object)]}]
        }]
    }
}
```

Figure 5: Prompt for question parser LLM(Part I).

```
Also, if you think there are no spatial area that needs to be specifically located, just simplify the format
as:
{
    "temporal_localizer": {
        "module": "temporal_truncate",
        "params": [{
            "module": "temporal_localize",
            "params": [$(a preposition), {"module": "detect_act", "params": [$(description of an event)]}]
        }]
    },
    "spatial_localizer": {
        "module": "spatial_truncate",
        "params": []
    }
}
If you think there are more than one area to be spatially located, you can add params to the
"spatial_truncate" module:
{
    "temporal_localizer": {
        "module": "temporal_truncate",
        "params": [{
            "module": "temporal_localize",
            "params": [$(a preposition), {"module": "detect_act", "params": [$(description of an event)]}]
        }]
    },
    "spatial_localizer": {
        "module": "spatial_truncate",
        "params": [{
            "module": "spatial_localize",
            "params": [$(a relationship word), {"module": "locate_obj", "params": [$(description of an
object A)]}]
        },
        {
            "module": "spatial_localize",
            "params": [$(a relationship word), {"module": "locate_obj", "params": [$(description of an
object B)]}]
        }]
    }
}
Except what I have mentioned above, you are not allowed to change the format of the policy in other
strange ways.
.
Here is some examples:
1.question: What is the boy holding after his mother entering the room?
policy:
{
    "temporal_localizer": {
        "module": "temporal_truncate",
        "params":[{
            "module": "temporal_localize",
            "params": ["after", {"module": "detect_act", "params": ["a woman enters the room"]}]
        }]
    },
    "spatial_localizer": {
        "module": "spatial_truncate",
        "params": [{
            "module": "spatial_localize",
            "params": ["holding", {"module": "locate_obj", "params": ["a boy"]}]
        }]
    }
}
```

Figure 6: Prompt for question parser LLM(Part II).

```
2.question: Why is the girl crying?
policy:
{
    "temporal_localizer": {
        "module": "temporal_truncate",
        "params":[{
            "module": "temporal_localize",
            "params": ["when", {"module": "detect_act", "params": ["a girl is crying"]}]
        }]
    },
    "spatial_localizer": {
        "module": "spatial_truncate",
        "params": [{
            "module": "spatial_localize",
            "params": ["surrounding", {"module": "locate_obj", "params": ["a girl"]}]
        }]
    }
}
3.question: What is the animal on the left of the farmer?
policy:
{
    "temporal_localizer": {
        "module": "temporal_truncate",
        "params":[]
    },
    "spatial_localizer": {
        "module": "spatial_truncate",
        "params": [{
            "module": "spatial_localize",
            "params": ["left of", {"module": "locate_obj", "params": ["a farmer"]}]
        }]
    }
}
4.question: Which object did the person put down before they took the sandwich?
policy:
{
    "temporal_localizer": {
        "module": "temporal_truncate",
        "params": [{
            "module": "temporal_localize",
            "params": ["before", {"module": "detect_act", "params": ["the person took the sandwich"]}]
        }]
    },
    "spatial_localizer": {
        "module": "spatial_truncate",
        "params": [{
            "module": "spatial_localize",
            "params": ["putting down", {"module": "locate_obj", "params": ["the person"]}]
        }]
    }
}
```
Expand your thinking, and don't be confined to the words and phrases in the original question. Remember that your goal in decomposing the question is to temporally and spatially locate it within the video. Therefore, feel free to think creatively about how breaking down the question might help find the parts of the video that are truly relevant to the question. Based on the original question, you can make some reasonable extensions and provide appropriate policies.
Note that for the "Why" type question, it is always not enough to just locate the event and the object mentioned in the origin question. You can try to think about what events and objects might be related to the question and try to locate them.
Now I will give you a question, and you will give me the corresponding policy.

Figure 7: Prompt for question parser LLM(Part III).

## A.2 MODULE IMPLEMENTATION

All small modules in our MoST-Grounding module are listed in Table 4, the detailed implementation of which are as follows:

`DetectAct`. Define the UniVTG model as $M_T$, text encoder as $E_t$, video encoder as $E_v$. $V = (v_1, v_2, ..., v_T$ is an input video with $T$ frames sampled at 1fps, and $Q_a$ is a query desribing an action. We have:

$$TS = M_T(E_v(V), E_t(Q_a)) \tag{7}$$

where $TS = [t_s, t_e]$ represents a time period.

`TemporalLocalize`. $p$ is a preposition representing certain temporal relationship. $TS^{(in)} = [t_s^{(in)}, t_e^{(in)}]$ is an input time span. We have:

$$TS^{(out)} = \begin{cases} TS^{(in)}, & p \in \{\text{when, while, as}\} \\ [t_e^{(in)}, \min(2t_e^{(in)} - t_s^{(in)}, T)], & p \in \{\text{after}\} \\ [\max(0, 2t_s^{(in)} - t_e^{(in)}), t_s^{(in)}], & p \in \{\text{before}\} \end{cases} \tag{8}$$

where $TS^{(out)} = [t_s^{(out)}, t_e^{(out)}]$ is an output time span. $T$ is the video duration.

`TemporalBetween`. Given two input time spans $TS_1 = [t_{1s}, t_{1e}]$ and $TS_2 = [t_{2s}, t_{2e}]$, we have:

$$TS^{(out)} = [\min(t_{1s}, t_{2s}), \max(t_{1e}, t_{2e})] \tag{9}$$

where $TS^{(out)} = [t_s^{(out)}, t_e^{(out)}]$ is an output time span.

`TemporalTruncate`. Given an input video $V = (v_1, v_2, ..., v_T)$ and a time span $TS = [t_s, t_e]$, define $s = \lfloor t_s \rfloor$, $e = \lceil t_e \rceil$. We get $I = (I_1, I_2, ...I_n)$, where:

$$I_i = v_{(s + \frac{(e-s)(i-1)}{n-1})} \tag{10}$$

and $n$ denotes the number of sampled frames.

`LocateObj`. Given the YOLO-World model $M_S$, an input image $I$, a query of an object $Q_o$, and an image encoder $E_I$ together with a text encoder $E_T$, we have:

$$BBOX = M_S(E_I(I), E_T(Q_o)) \tag{11}$$

where $BBOX = (x_1, y_1, x_2, y_2)$ is an output bounding box.

`SpatialLocalize`. Given an input image $I$, an input bounding box $BBOX^{(in)} = (x_1, y_1, x_2, y_2)$ and a word or phrase representing certain spatial relationship $r$, we have:

$$BBOX^{(out)} = \begin{cases} [\max(0, 2x_1 - x_2), y_1, x_1, y_2], & p \in \{\text{left}\} \\ [x_2, y_1, \min(w, 2x_2 - x_1), y_2], & p \in \{\text{right}\} \\ [x_1, y_2, x_2, \min(h, 2y_2 - y_1)], & p \in S_{\text{down}} \\ [x_1, \max(0, 2y_1 - y_2), x_2, y_1], & p \in S_{\text{up}} \\ [\max(0, 2x_1 - x_2), \max(0, 2y_1 - y_2), \\ \min(w, 2x_2 - x_1), \min(h, 2y_2 - y_1)], & p \in S_{\text{surround}} \end{cases} \tag{12}$$

where $S_{\text{down}} = \{\text{bottom, down, below, under, beneath, sit on, stand on, lying on}\}$, $S_{\text{up}} = \{\text{top, above, up, carry, lift, on}\}$, $S_{\text{surround}} = \{\text{next to, beside, near, surround}\}$. Particularly, if $p \in \{\text{hold, touch, contact, take}\}$, we have:

$$BBOX^{(out)} = \texttt{SpatialLocalize}(I, BBOX^{(\text{hand})}, \text{"surround"}) \tag{13}$$

where $BBOX^{(\text{hand})} = \texttt{LocateObj}(I, \text{"hand"})$.

`SpatialTruncate`. Given an input image I and a list of bounding boxes $BBOX_1, BBOX_2, ...$ where $BBOX_i = (x_{i1}, y_{i1}, x_{i2}, y_{i2})$, we have:

$$I^{(out)} = RESIZE_I(I[\min_i x_{i1}, \min_i y_{i1}, \max_i x_{i2}, \max_i y_{i2}]) \tag{14}$$

where $RESIZE_I(I')$ is the operation that resizes an image $I'$ into the shape of $I$.

## A.3 MULTIMODAL LLM INPUT FORMAT

The specific input of our multimodal LLM is illustrated in Figure 8 together with our instruction prompt. Global representation tokens are encoded and aligned global video representation $g_v$. Similarly, temporal-grounded video tokens and spatial-grounded video tokens are encoded and aligned video segments $v_s$ and bounding boxes $b_{v_s}$ respectively. Special tokens glob, tp and sp are designed to help the multimodal LLM understand different types of tokens.

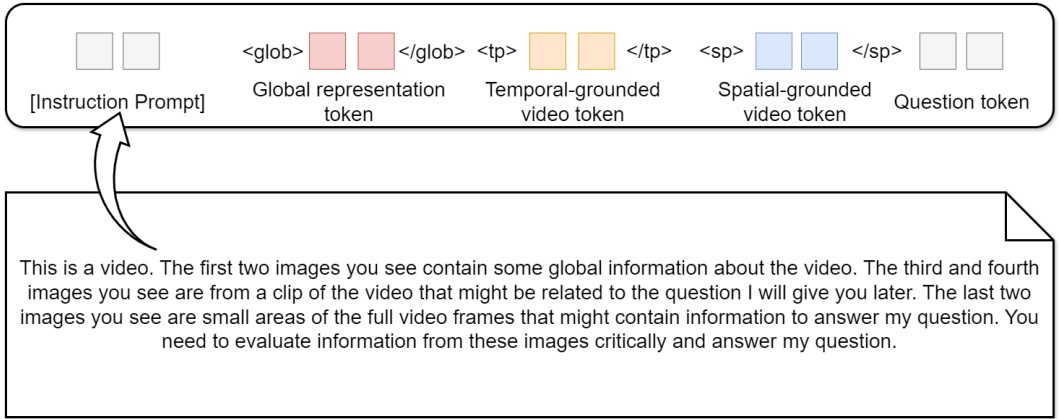

Figure 8: Specific input of our multimodal LLM and the instruction prompt.

## A.4 DETAILED ALTERNATE SELF-REFLECTION TRAINING STRATEGY

The detail of our Alternate Self-reflection Training Strategy is demonstrated in Algorithm 1.

## A.5 MORE EXAMPLES

Here we present some more inference examples of our MSR-ViR framework.

**Q: Why does the child pass the toy back to the person in black after holding it and walking?**

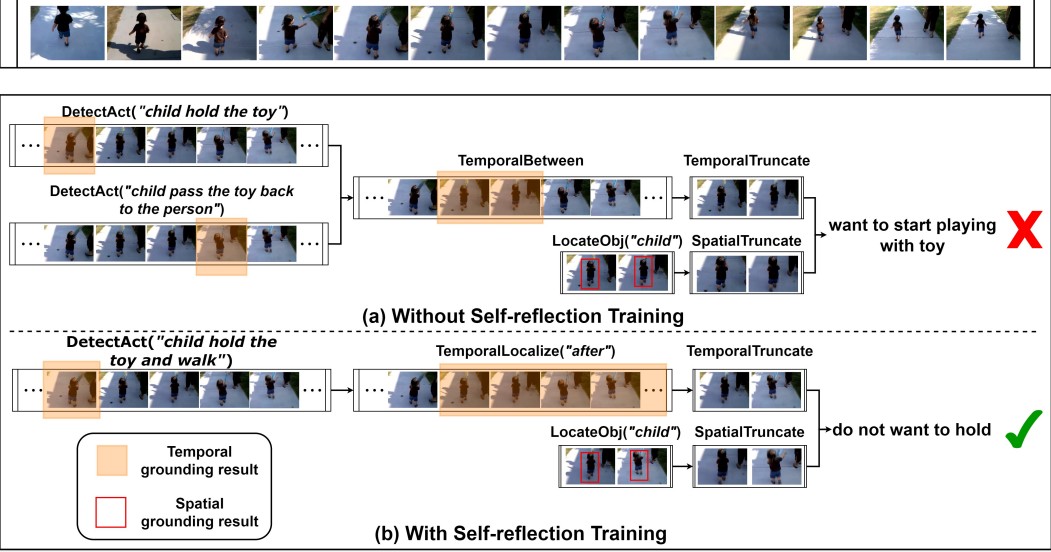

Figure 9: MSR-ViR inference example 1.

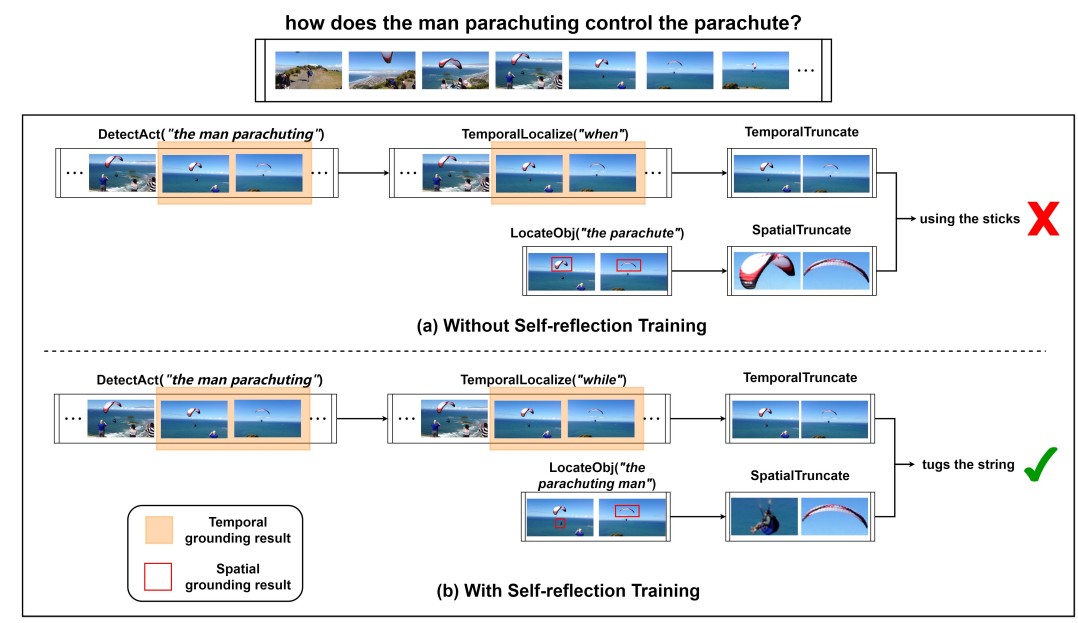

Figure 10: MSR-ViR inference example 2.

---

**Algorithm 1** Alternate Self-reflection Training Strategy

1: **Input:** Question parser LLM $Q$, MoST-Grounding module $\mathcal{M}$, **temporal localizer** $\mathcal{F}_t$, **spatial localizer** $\mathcal{F}_s$, modules in **temporal localizer** $\{m_i^t\}_{i=1}^4$, modules in **spatial localizer** $\{m_i^s\}_{i=1}^3$, multimodal LLM $\mathcal{F}$, instruction prompt $Pt$, dataset $D = \{(v_i, q_i, y_i)\}_{i=1}^N$, total training steps $S$, gradient accumulate step $s$, alternate training period $P$
2: **Initialize:** $Q, \mathcal{M}, CACHE$
3: **Freeze:** $\mathcal{M}, \{m_i^t\}, \{m_i^s\}, Q$, **Activate:** $\mathcal{F}$
4: **for** $t = 1, \ldots, S$ **do**
5:     **for** $j = 1, \ldots, s$ **do**
6:         $i \leftarrow ((t-1)s + j - 1)\%N + 1$, Prepare data $(v_i, q_i, y_i)$, Derive global representation $g_v$.
7:         Generate policy $p = Q(q_i)$
8:         Set $\mathcal{F}_{t|p}(\cdot) = \mathcal{I}(\{m_i^t\}, p)(\cdot), \mathcal{F}_{s|p}(\cdot) = \mathcal{I}(\{m_i^s\}, p)(\cdot)$, derive $c_t, c_s$ from $p$
9:         $\mathcal{M}$ **execution:** $v_s, b_{v_s} = \mathcal{M}(v_i, c_t, c_s) = \mathcal{F}_s\big(\mathcal{F}_t(v, c_t), c_s\big)$
10:        $\mathcal{F}$ **forward propagation:** $\hat{y}(q_i, v_i) = \mathcal{F}\big(Pt(q_i, v_s, b_{v_s}, g_v)\big)$
11:        **Optimize** $\mathcal{F}$ **with loss:** $\mathcal{L}_{\text{CE}}$ in Equation (5)
12:        Add $(v_i, q_i, y_i)$ to $CACHE$
13:     **end for**
14:     **if** $t\%P = 0$ **then**
15:        Freeze $\mathcal{F}$, activate $Q$, initialize $\pi_\theta = Q, \pi_{\text{ref}} = Q$
16:        **for** $i = 1, \ldots, sP$ **do**
17:            Prepare data $(v_i, q_i, y_i)$
18:            Generate policies $p_1, p_2 = Q(q_i)$
19:            Forward propagation to get $\mathcal{L}_{\text{CE}1}, \mathcal{L}_{\text{CE}2}$ for $p_1, p_2$ respectively
20:            **if** $\mathcal{L}_{\text{CE}1} < \mathcal{L}_{\text{CE}2}$ **then**
21:               $p_w \leftarrow p_1, p_l \leftarrow p_2$
22:            **else**
23:               $p_w \leftarrow p_2, p_l \leftarrow p_1$
24:            **end if**
25:            **Optimize** $\pi_\theta$ **with loss:** $\mathcal{L}_{\text{DPO}}$ in Equation (6)
26:        **end for**
27:        $Q \leftarrow \pi_\theta$, **clear** $CACHE$, freeze $Q$, activate $\mathcal{F}$
28:     **end if**
29: **end for**

## A.6 FURTHER ABLATION STUDIES

In MoST-Grounding module, we utilize a small grounding model UniVTG as our temporal grounding module. To demonstrate the effectiveness of UniVTG, we further conduct ablation study on the choice of temporal grounding model. We utilize $R^2$-Tuning(Liu et al., 2024) and Moment-DETR(Lei et al., 2021) to replace UniVTG and test on NExT-GQA dataset, and the results are shown in Table 5. MSR-ViR$_Q$ with UniVTG achieves the best results on NExT-GQA.

Table 5: Ablation study for temporal grounding models on NExT-GQA. This is the test result of MSR-ViR$_Q$ with different temporal grounding models UniVTG, $R^2$-Tuning and Moment-DETR.

| Grounding Model | Acc@QA | Acc@GQA | mIoP | IoP@0.5 | mIoU | IoU@0.5 |
|---|---|---|---|---|---|---|
| UniVTG | **69.9** | **18.5** | **30.0** | **25.0** | **22.8** | **16.4** |
| $R^2$-Tuning | 67.3 | 16.6 | 28.7 | 23.2 | 22.7 | 15.9 |
| Moment-DETR | 67.4 | 17.2 | 28.6 | 24.1 | 21.4 | 14.7 |

For further comparison between end-to-end Multimodal LLMs and our MSR-ViR, we provide their parameter count and inference speed in Table 6. The inference speed is tested on two NVIDIA A100 GPUs. For MSR-ViR$_Q$, the parameter size is: Qwen2(7B) + Qwen-VL(9.6B) + YOLO-World(48M) + UniVTG(41.3M) $\approx$ 16.6B.

Table 6: Parameter size and inference speed of Qwen-VL and MSR-ViR$_Q$

| | Parameter Size | Inference Speed |
|---|---|---|
| Qwen-VL | 9.6B | 1.29 qa paris / s |
| MSR-ViR$_Q$(ours) | 16.6B | 0.21 qa pairs / s |

