# OpenReview forum: "MSR-ViR: Modularized Self-reflected Video Reasoner for Video Question Answering"
_ICLR.cc/2025/Conference — Submitted to ICLR 2025_

### Official Review · Reviewer_1xgn · 2024-10-20

**Soundness:** 3
**Presentation:** 3
**Contribution:** 3
**Rating:** 6
**Confidence:** 3

**Summary:**

This paper addresses the black-box problem of multimodal large language models (MLLMs) in VideoQA by proposing MSR-ViR, a novel framework. MSR-ViR introduces two core components: (1) the MoST-Grounding module, which localizes relevant temporal segments and spatial regions in videos, and (2) an Alternate Self-Reflection Training Strategy, which iteratively enhances the question parser and the MLLM. Evaluations on datasets such as NExT-QA, STAR, and NExT-GQA demonstrate that MSR-ViR achieves competitive performance on VideoQA tasks and improves interpretability by providing visual evidence for answers.

**Strengths:**

1) The paper is well written and the motivation is clear, with a strong focus on the interpretability challenge in VideoQA, making it highly relevant to the field.
2) The MoST-Grounding module integrates multiple submodules to effectively localize temporal and spatial regions, improving transparency in the input to the MLLM.
3) The Alternate Self-Reflection strategy introduces a novel reinforcement-based method to align the question parser and the MLLM, enhancing performance and consistency.

**Weaknesses:**

1) The framework is relatively heavy, relying on multiple external tools for tasks and additional operations such as resizing and truncation, which increases computational overhead. Moreover, what if these external tools are unreliable, which can lead to further exposure bias? It's necessary to further investigate the choice of the sub-modules in the MoST-Grounding module.
2) While the approach improves the selection of input information, it does not make the internal reasoning process of the MLLM more interpretable. It still focuses on the process of 'input' to decide which information should be fed into the MLLM as soft prompts.
3) The paper misses references with related works such as SeViLa [1] and GCG [2], which also focus on VideoQA with grounding elements. Including these baselines would strengthen the empirical validation.

[1] Yu et al. "Self-Chained Image-Language Model for Video Localization and Question Answering", 2023 NIPS
[2] Wang et al. "Weakly Supervised Gaussian Contrastive Grounding with Large Multimodal Models for Video Question Answering", 2024 ACM MM

**Questions:**

See Weaknesses

---

> ### Author Response · Authors · 2024-11-24
> **Responses to Reviewer 1xgn (Part I)**
>
> We sincerely thank the reviewer for taking time to review our paper and providing thoughtful feedback and insightful suggestions. We address the weaknesses as follows:
>
> **Weakness1**
>
> The MoST-Grounding module introduces a temporal grounding tool, UniVTG, and a spatial grounding tool, YOLO-World. These two models are relatively small compared to our base LLM, with UniVTG having 41.3M parameters and YOLO-World 48M parameters. Additionally, both models have fast inference speeds, so the MoST-Grounding module does not introduce significant computational costs. In fact, the primary source of computational overhead in our framework is the question parser based on Qwen2-7B, which is quite common in VideoQA tasks based on LLMs. Many reasoning-based VideoQA models(like MoReVQA[1], LLoVi[2] and VideoTree[3]) rely on LLMs, which inevitably introduce additional computational overhead.
>
> When selecting the small modules for the MoST-Grounding module, we consider not only the models’ ability to perform grounding tasks but also their operational efficiency, as highlighted by the reviewer. If the small modules were slow, they would substantially affect the framework's overall efficiency. The chosen UniVTG and YOLO-World models strike a balance between strong grounding capabilities and high efficiency.
>
> Following the reviewer’s suggestion, we replace UniVTG with two temporal grounding models, $R^2$-Tuning[4] and Moment-DETR[5], on the NExT-GQA dataset as a further ablation study. The results are presented in the table below. As shown, both models exhibit lower temporal grounding accuracy compared to UniVTG (especially in the IoP metric), leading to reduced VideoQA accuracy and Acc@GQA for the MSR-ViR framework.
>
> |                           | Acc@QA   | Acc@GQA  | mIoP     | IoP@0.5  | mIoU     | IoU@0.5  |
> | ------------------------- | -------- | -------- | -------- | -------- | -------- | -------- |
> | MSR-ViR (w/ UniVTG)       | **69.9** | **18.5** | **30.0** | **25.0** | **22.8** | **16.4** |
> | MSR-ViR (w/ $R^2$-Tuning) | 67.3     | 16.6     | 28.7     | 23.2     | 22.7     | 15.9     |
> | MSR-ViR (w/ Moment-DETR)  | 67.4     | 17.2     | 28.6     | 24.1     | 21.4     | 14.7     |
>
> Since spatial grounding must be performed for every temporal-grounded frame, the efficiency requirements for the spatial grounding model are even higher. YOLO-World’s exceptional efficiency makes it particularly well-suited to our needs, offering both high efficiency and high grounding accuracy—features that other open-vocabulary object detector models lack[6].
>
> [1] Min, Juhong, et al. "MoReVQA: Exploring Modular Reasoning Models for Video Question Answering." *Proceedings of the IEEE/CVF Conference on Computer Vision and Pattern Recognition*. 2024.
>
> [2] Zhang, Ce, et al. "A simple llm framework for long-range video question-answering." *arXiv preprint arXiv:2312.17235* (2023).
>
> [3] Wang, Ziyang, et al. "VideoTree: Adaptive Tree-based Video Representation for LLM Reasoning on Long Videos." *arXiv preprint arXiv:2405.19209* (2024).
>
> [4] Liu, Ye, et al. "$ R^ 2$-Tuning: Efficient Image-to-Video Transfer Learning for Video Temporal Grounding." *arXiv preprint arXiv:2404.00801* (2024).
>
> [5] Lei, Jie, Tamara L. Berg, and Mohit Bansal. "Detecting moments and highlights in videos via natural language queries." *Advances in Neural Information Processing Systems* 34 (2021): 11846-11858.
>
> [6] Cheng, Tianheng, et al. "Yolo-world: Real-time open-vocabulary object detection." *Proceedings of the IEEE/CVF Conference on Computer Vision and Pattern Recognition*. 2024.

---

> ### Author Response · Authors · 2024-11-24
> **Responses to Reviewer 1xgn (Part II)**
>
> **Weakness2**
>
> In our work, the interpretability we emphasize refers to clearly demonstrating how, step by step, the relevant temporal segments are located from the video based on a given question, and how the spatial regions relevant to the question are then inferred from the frames extracted from these temporal segments. The interpretability mentioned by the reviewer refers to the process of step-by-step reasoning from visual information and the question to the final answer. These represent two different definitions of interpretability and can also be seen as two essential steps in completing the VideoQA task: first, locating the temporal and spatial segments, and then reasoning the answer based on the localized content. Our work focuses on the former. In future work, we will incorporating the internal reasoning process of the LLM.
>
> **Weakness3**
>
> Based on the reviewer’s suggestion, we introduced SeViLa’s test results on NExT-QA, STAR, and NExT-GQA, as well as GCG’s test results on NExT-QA for comparison with the Llava-Next version of our MSR-ViR framework. The results, presented in the table below, demonstrate a significant improvement in VideoQA accuracy, surpassing existing grounding-based methods such as SeViLa and GCG.
>
> |                                 | NExT-QA Tem. | NExT-QA Cau. | NExT-QA Des. | NExT-QA Avg. | STAR-sub   Int. | STAR-sub Seq. | STAR-sub Avg. |
> | ------------------------------- | ------------ | ------------ | ------------ | ------------ | --------------- | ------------- | ------------- |
> | SeViLa                          | 69.4         | 74.2         | **81.3**     | 73.8         | 63.7            | 70.4          | 67.1          |
> | GCG                             | **72.6**     | 74.2         | 80.7         | 74.6         | -               | -             | -             |
> | MSR-ViR(ours, Llava-Next-based) | 72.2         | **74.6**     | 80.9         | **74.9**     | **68.9**        | **73.1**      | **71.0**      |
>
> |                                 | Acc@GQA  | mIoP     | IoP@0.3  | IoP@0.5  | mIoU     | IoU@0.3  | IoU@0.5  |
> | ------------------------------- | -------- | -------- | -------- | -------- | -------- | -------- | -------- |
> | SeViLa                          | 16.6     | 29.5     | 34.7     | 22.9     | 21.7     | 29.2     | 13.8     |
> | MSR-ViR(ours, Llava-Next-based) | **18.6** | **29.6** | **39.0** | **24.1** | **23.4** | **33.6** | **16.4** |

---

> ### Author Response · Authors · 2024-11-27
> **A gentle reminder**
>
> Dear reviewer, thank you again for the review and we hope that our response and the uploaded revised paper have addressed your concerns. We would greatly appreciate your feedback and please feel free to let us know if you have any other questions.

---

> > ### Comment · Reviewer_1xgn · 2024-11-29
> >
> > Thanks for your efforts in addressing my concerns, and I hope the authors will add these experiments and analyses to the final version to make it more comprehensive for readers. In conclusion, I will raise my score to 6.

---

> ### Author Response · Authors · 2024-11-29
>
> We sincerely thank the reviewer for the insightful feedback and suggestions. The analyses and experiments according to the reviewer's suggestions further improve the quality of our paper, which have been added to the revised version of the paper.

---

### Official Review · Reviewer_P6dF · 2024-10-29

**Soundness:** 3
**Presentation:** 3
**Contribution:** 2
**Rating:** 6
**Confidence:** 4

**Summary:**

This paper aims to improve the interpretability of Multimodal LLMs in performing VideoQA. To achieve the goal, the authors design a modular self-reflection framework MSR-ViR. The framework primarily comprises a spatial-temporal grounding module and a self-refection learning mechanism based on DPO. MSR-ViR basically decouples video grounding from videoqa, enabling the interpretaion of intermediate results to understand the answers. The experiments on related datasets have demonstrated the strength of the approach in both accuracy and interpretability (grounded accuracy).

**Strengths:**

1.	Successfully develop a ground-then-answer framework for interpretable video question parsing. The question parser policy is able to be optimized via answer feedback.

2.	The approach is well presented and easy to understand.

**Weaknesses:**

1.	The paper does not improve video grounding but just uses existing method UniVTG. According to table 2, the grounding performance in terms of IoP@0.5 is worse than previous VLMs (VGT and TempCLIP). This severely limits the improvements of QA performance.
2.	According to the model ablation results in Table 3,  the global representation g_v (which opens back door for grounded QA) seems more crucial than other components. Such results slightly depart from the major claim of interpretable VQA where correct answers are anchored on correct visual content.
3.	Should compare with SeViLA which also finetunes a localizer on QVHighlight (like UniVTG) for grounded VideoQA.

**Questions:**

see weaknesses.

---

> ### Author Response · Authors · 2024-11-24
> **Responses to Reviewer P6dF**
>
> We sincerely thank the reviewer for taking time to review our paper and providing thoughtful feedback and insightful suggestions. We address the weaknesses as follows:
>
> **Weakness1**
>
> We use the UniVTG model, pretrained on several temporal grounding datasets, as the temporal localization tool in the MoST-Grounding module. However, we have not conducted further training for UniVTG on our dataset, NExT-GQA. As shown in Table 2, while our temporal grounding results are slightly weaker than Temp[CLIP] and VGT in the IoP@0.5 metric, our performance is significantly stronger across all other grounding metrics. We believe this demonstrates that MSR-ViR with the pretrained temporal grounding module UniVTG has superior temporal localization capabilities.
>
> **Weakness2**
>
> Since the temporal grounding module cannot achieve 100% accuracy in identifying the time segments relevant to the question, sampling video frames solely from the grounding results may still result in missing some information. Therefore, the global representation $g_v$ serves as a necessary "back door". However, $g_v$ is not the most critical source of information—frames sampled from the temporal grounding results and spatial grounding results are more important. To demonstrate this, we have supplemented the ablation study on NExT-QA by removing the temporal grounding frames and spatial grounding frames, leaving only the global representation $g_v$. The results show a significant drop in question-answering accuracy, proving the importance of the grounded frames.
>
> |                                  | Tem.     | Cau.     | Des.     | Avg.     |
> | -------------------------------- | -------- | -------- | -------- | -------- |
> | MSR-ViR                          | 69.9     | 73.4     | 81.5     | 73.6     |
> | MSR-ViR(w/o self-reflection)     | 67.2     | 72.5     | 80.5     | 72.1     |
> | MSR-ViR(w/o $g_v$)               | 66.9     | 70.1     | 78.0     | 70.4     |
> | MSR-ViR(w/o instruction prompts) | 68.3     | 72.4     | 82.4     | 72.8     |
> | MSR-ViR(w/o spatial modules)     | 67.0     | 72.5     | 81.4     | 72.2     |
> | **MSR-ViR (only w/ $g_v$)**      | **63.3** | **68.4** | **77.5** | **68.3** |
>
> **Weakness3**
>
> Based on the reviewer’s suggestion, we introduced SeViLa’s testing results on NExT-QA, STAR, and NExT-GQA for comparison with the Llava-Next version of our MSR-ViR framework.
>
> |                                 | NExT-QA Tem. | NExT-QA Cau. | NExT-QA Des. | NExT-QA Avg. | STAR-sub   Int. | STAR-sub Seq. | STAR-sub Avg. |
> | ------------------------------- | ------------ | ------------ | ------------ | ------------ | --------------- | ------------- | ------------- |
> | SeViLa                          | 69.4         | 74.2         | **81.3**     | 73.8         | 63.7            | 70.4          | 67.1          |
> | MSR-ViR(ours, Llava-Next-based) | **72.2**     | **74.6**     | 80.9         | **74.9**     | **68.9**        | **73.1**      | **71.0**      |
>
> |                                 | Acc@GQA  | mIoP     | IoP@0.3  | IoP@0.5  | mIoU     | IoU@0.3  | IoU@0.5  |
> | ------------------------------- | -------- | -------- | -------- | -------- | -------- | -------- | -------- |
> | SeViLa                          | 16.6     | 29.5     | 34.7     | 22.9     | 21.7     | 29.2     | 13.8     |
> | MSR-ViR(ours, Llava-Next-based) | **18.6** | **29.6** | **39.0** | **24.1** | **23.4** | **33.6** | **16.4** |

---

> ### Author Response · Authors · 2024-11-27
> **A gentle reminder**
>
> Dear reviewer, thank you again for the review and we hope that our response and the uploaded revised paper have addressed your concerns. We would greatly appreciate your feedback and please feel free to let us know if you have any other questions.

---

> > ### Comment · Reviewer_P6dF · 2024-11-27
> >
> > Thank you for the responses. My concerns are addressed and I increase the score to 6.

---

> ### Author Response · Authors · 2024-11-27
>
> We sincerely thank the reviewer for the response and feedback. The insightful and constructive suggestions further improve the quality of our paper.

---

### Official Review · Reviewer_ndrP · 2024-11-01

**Soundness:** 4
**Presentation:** 3
**Contribution:** 3
**Rating:** 3
**Confidence:** 5

**Summary:**

The paper addresses the use of multimodal Large Language Models in understanding tasks across various multimodal scenarios, specifically focusing on applications in Video Question Answering. However, current Multimodal Large Language Models are largely black-box systems for VideoQA tasks, lacking the ability to provide an understandable reasoning path and, thus, suffer from limited interpretability. To address this, the authors propose MSR-ViR, which constructs a modular reasoning structure designed to generate reasoning paths, and incorporates a reinforcement learning framework to prevent the model from generating unreasonable reasoning paths.

While the proposed approach is interesting, it relies on the integration of four existing models. This ensemble-based structure shows only marginal performance improvements (1-2%), and the manuscript does not discuss crucial aspects such as reasoning time costs or memory overhead.

**Strengths:**

1. The MoST-Grounding module introduces two standard modules—temporal localizer and spatial localizer—which can be flexibly assembled in sequence based on the question parser. This structure is robust and allows for reliable generation of reasoning paths.

2. The authors present a clear motivation: to create a framework for generating reasoning paths for the black-box nature of VideoQA tasks. The comprehensive visualization of reasoning paths further demonstrates the effectiveness of the model.

**Weaknesses:**

1. The question parser and LLM reasoning components both rely on LLM structures, leading to high computational costs.

2. Both the temporal localizer and spatial localizer use existing models, specifically UniVTG and YOLO-World, which contribute to significant parameter overhead. As the complexity of VideoQA tasks increases, relying on these two models alone may not only limit predictive accuracy but also compromise the completeness of reasoning paths. Future work may need to explore additional modules to support diverse combinations (see [1] for reference).

3. The ablation study lacks comprehensiveness. While the authors assess model performance on QA and grounding tasks and provide an effectiveness analysis of each module, they do not evaluate inference speed, parameter count, or other metrics compared to end-to-end models. Given that the proposed framework integrates multiple existing large models, an analysis of inference speed is both important and currently missing.

[1]. Neural Module Networks

**Questions:**

To generate reasoning paths for VideoQA, would it be more effective to design a Chain-of-Thought dataset and perform supervised fine-tuning (SFT)? The O1 model currently adopts this approach, achieving clear reasoning paths through an end-to-end structure.

---

> ### Author Response · Authors · 2024-11-24
> **Responses to Reviewer ndrP**
>
> We sincerely thank the reviewer for taking time to review our paper and providing thoughtful feedback and insightful suggestions. We address the weaknesses and questions as follows:
>
> **Weakness 1**
>
> Our MSR-ViR framework employs Qwen2-7B as the question parser and Qwen-VL as the multimodal answerer, both of which are based on LLM architectures. However, many recent works utilize LLMs to tackle VideoQA tasks. For instance, MoReVQA[1] leverages Palm-2, which is also a large language model, for four times in order to reason and get the answer to one question. Reasoning tasks often require step-by-step reasoning to derive multiple intermediate results before arriving at the final answer, which distinguishes them from end-to-end models that directly predict the final answer. This difference also explains why reasoning-based approaches typically cost greater computational overhead.
>
> **Weakness 2**
>
> To enhance the interpretability of the VideoQA task, the MoST-Grounding module employs UniVTG as the temporal grounding model and YOLO-World as the spatial grounding model, which indeed introduces additional parameters. However, both grounding models have relatively small parameter sizes: UniVTG has 41.3M parameters, and YOLO-World has 48M parameters, which are only a tiny fraction of the 7B parameters in LLMs. Therefore, the MoST-Grounding module does not significantly increase the overall parameter count of the framework.
>
> The MoST-Grounding (Modularized Spatial-Temporal Grounding) module serves to localize the original question temporally and spatially within the video, providing the Multimodal LLM with the most relevant information to the question. We sincerely appreciate the reviewer’s valuable suggestion, as it is true that the current modular network employs only one temporal grounding model and one spatial grounding model. While experiments have already demonstrated satisfactory localization performance (see Table 2), incorporating more modules into the modular network to enhance its functionality could potentially further improve localization and answer accuracy. We will try to introduce more modules into our framework in future work.
>
> **Weakness 3**
>
> Based on the reviewer’s suggestion, we provide experimental results on inference speed and parameter size as follows:
>
> |               | Parameter Size                           | Inference Speed   |
> | ------------- | ---------------------------------------- | ----------------- |
> | Qwen-VL       | 9.6B                                     | 1.29 qa pairs / s |
> | MSR-ViR(ours) | qwen2(7B) + qwen-vl(9.6B) + yolo-world(48M) + univtg(41.3M) = 16.6B | 0.21 qa pairs / s |
>
> The inference speed results were tested on two NVIDIA A100 GPUs. As shown, MSR-ViR introduces a larger parameter size compared to end-to-end Multimodal LLM. Additionally, since MSR-ViR requires an LLM to first generate a policy, followed by the MoST-Grounding modular network for grounding, and finally the multimodal answerer to provide the answer, its inference speed is noticeably slower than that of end-to-end Multimodal LLM.
>
> Actually, the additional time cost is normal for reasoning tasks. As tested and reported in [2], the inference speed of VideoTree is about 0.13 qa pairs / s, while the inference speed of LLoVi[3] is about 0.26 qa pairs / s. Similarly, GPT-O1 takes significantly longer time to answer questions than GPT-4o because it performs detailed reasoning process while generating responses.
>
> **Question 1**
>
> We sincerely thank the reviewer for providing this insightful suggestion! In our current work, we leverage feedback from the Multimodal LLM as a reward to guide the training of the question parser through reinforcement learning. As shown in the ablation study in Table 3, this reinforcement learning approach improves the ability of the question parser to generate reasonable policies. However, reinforcement learning training may not be as stable or effective as direct supervised learning. If a Chain-of-Thought dataset could be constructed to enable supervised training of the question parser, it might further enhance its policy generation capability, thereby improving the overall performance of our framework. We sincerely apologize that, due to time constraints, we are currently unable to construct a high-quality CoT dataset and conduct training. We will explore this direction in our future work.
>
> [1] Min, Juhong, et al. "MoReVQA: Exploring Modular Reasoning Models for Video Question Answering." *Proceedings of the IEEE/CVF Conference on Computer Vision and Pattern Recognition*. 2024.
>
> [2] Wang, Ziyang, et al. "VideoTree: Adaptive Tree-based Video Representation for LLM Reasoning on Long Videos." *arXiv preprint arXiv:2405.19209* (2024).
>
> [3] Zhang, Ce, et al. "A simple llm framework for long-range video question-answering." *arXiv preprint arXiv:2312.17235* (2023).

---

> > ### Comment · Reviewer_ndrP · 2024-12-02
> > **Response to the Authors**
> >
> > Thank you for the reviewer’s response to my questions. The issues I was particularly concerned with, namely the novelty and inference time, have been addressed in detail:
> >
> > On the use of multiple existing models in MSR-ViR for ensembling: The author cites MoReVQA as an example. Although I am generally not fond of model ensembling, it is indeed a common practice in current submissions, which has convinced me on this point.
> >
> > On the inference time overhead of MSR-ViR: The table provided by the author shows that their inference speed is 6 times slower than QWen-VL. I believe that using self-reflection might further slow it down. While the author compares the inference speed with models like VideoTree and LLoVi, both of which are designed for long videos and thus present more challenging tasks, these models are not directly comparable (even though the author evaluates their model on NextQA, the lack of evaluation on longer datasets such as EgoSchema makes it unsuitable as a benchmark for long video LLMs). Furthermore, the author incurs a 6x time complexity increase but only shows a marginal improvement over QWen-VL in Table 1, making this inference time overhead intolerable.
> >
> > Therefore, I have decided to lower the score.

---

> ### Author Response · Authors · 2024-11-27
> **A gentle reminder**
>
> Dear reviewer, thank you again for the review and we hope that our response and the uploaded revised paper have addressed your concerns. We would greatly appreciate your feedback and please feel free to let us know if you have any other questions.

---

> ### Author Response · Authors · 2024-12-01
> **A gentle reminder**
>
> Dear reviewer, thank you again for the review and the insightful feedback. We hope that our response and the revised paper have addressed your concerns. As the discussion stage is about to end, we would greatly appreciate your feedback and please feel free to let us know if you have any other questions.

---

> ### Author Response · Authors · 2024-12-02
> **Further Response to Reviewer ndrP**
>
> We thank the reviewer again for the further feedback on our responses. Below, we provide further responses to the reviewer’s concerns:
>
> **1. Regarding the concern about inference speed:**
> We would like to emphasize again that reasoning-based inference and end-to-end inference are fundamentally different. It is entirely normal and common for reasoning-based inference to be significantly slower than end-to-end inference, as it not only provides answers but also generates step-by-step reasoning paths to deliver more accurate and interpretable responses.
>
> Still taking MoReVQA[1] as an example, this work shares similarities with ours, solving reasoning-based VideoQA tasks. Its framework includes four LLMs, two additional vision-language models and a video captioning model, with the reasoning process spanning four stages. Clearly, the computational cost of the MoReVQA framework is far greater than that of our MSR-ViR framework. Although the MoReVQA paper does not report inference speed and the code is not open-sourced, it is foreseeable that its inference speed is not significantly faster than our framework. Compared to its direct baseline JCEF, MoReVQA introduces considerable computational complexity and time overhead for its multi-stage reasoning process. However, it achieves a 2.5% accuracy improvement on NExT-QA, which is comparable to our MSR-ViR framework(see the table below). Just like MoReVQA and our MSR-ViR, most reasoning methods share similar characteristics, with increased computational costs compared to their base models. Contributions of reasoning methods extend beyond improving question-answering accuracy. Providing interpretable reasoning paths(and also evidence of answers in videos like MSR-ViR) is also a critical contribution, which inevitably brings greater computational overhead.
>
> It is worth noting that grounding-based methods also often have slower inference speeds. For example, SeViLa[2] has an inference speed of only 0.30 qa pairs / s(as mentioned in SeViLa paper's appendix), which is comparable to the inference speed of our MSR-ViR framework. Note that SeViLa is not designed for long-form videos.
>
> **2. Regarding the concern that the accuracy improvement of MSR-ViR on VideoQA datasets is marginal:**
> On the STAR-sub dataset, MSR-ViR(Qwen-VL-based) achieved a 3.4% improvement over its baseline (63.0 → 66.4). Specifically, on the Interaction and Sequence subsets, where temporal and spatial relationships are critical, it improved by 4.4% and 2.5%, respectively. We believe this significant accuracy improvement demonstrates the effectiveness of the MSR-ViR framework and is far from being marginal.
>
> On the NExT-QA dataset, as shown in the table below, we compared the performance improvement of our method and other grounding-based methods and modular methods relative to their respective base models. For example:
>
> - SeViLa improved by 1.2% over its base model BLIP-2$^{\text{concat}}$.
> - MoReVQA improved by 2.5% over its direct baseline JCEF.
> - Our MSR-ViR(Qwen-VL-based) improved by 1.7% over its base model Qwen-VL.
> - Our MSR-ViR(Llava-Next-based) improved by 1.8% over its base model Llava-Next.
>
> The improvements achieved by our method are comparable to those of existing grounding-based methods and modular methods, which we believe are not marginal.
>
> |                                 | NExT-QA Tem. | NExT-QA Cau. | NExT-QA Des. | NExT-QA Avg. |
> | ------------------------------- | ------------ | ------------ | ------------ | ------------ |
> | BLIP-2$^{\text{concat}}$        | 68.1         | 72.9         | 81.2         | 72.6         |
> | SeViLa                          | 69.4(+1.3)   | 74.2(+1.3)   | 81.3(+0.1)   | 73.8(+1.2)   |
> | JCEF                            | 61.6         | 68.3         | -            | 66.7         |
> | MoReVQA                         | 64.6(+3)     | 70.2(+1.9)   | -            | 69.2(+2.5)   |
> | Qwen-VL                         | 68.4         | 71.3         | 80.6         | 71.9         |
> | MSR-ViR(ours, Qwen-VL-based)    | 69.9(+1.5)   | 73.4(+2.1)   | 81.5(+0.9)   | 73.6(+1.7)   |
> | Llava-Next                      | 69.5         | 73.3         | 79.7         | 73.1         |
> | MSR-ViR(ours, Llava-Next-based) | 72.2(+2.7)   | 74.6(+1.3)   | 80.9(+1.2)   | 74.9(+1.8)   |
>
> [1] Min, Juhong, et al. "MoReVQA: Exploring Modular Reasoning Models for Video Question Answering." *Proceedings of the IEEE/CVF Conference on Computer Vision and Pattern Recognition*. 2024.
>
> [2] Yu, Shoubin, et al. "Self-chained image-language model for video localization and question answering." *Advances in Neural Information Processing Systems* 36 (2024).

---

### Official Review · Reviewer_gPWW · 2024-11-19

**Soundness:** 3
**Presentation:** 3
**Contribution:** 3
**Rating:** 5
**Confidence:** 2

**Summary:**

This paper presents MSR-ViR (Modularized Self-Reflected Video Reasoner), a framework for Video Question Answering (VideoQA) that enhances interpretability by integrating multimodal large language models (LLMs) with spatial-temporal grounding and self-reflective training. Traditional multimodal LLMs struggle with interpretability, as they operate as black-box systems without revealing the reasoning process or the video segments informing their answers. MSR-ViR addresses this by using a MoST-Grounding module to localize relevant video segments and spatial regions based on policies generated by a question-parsing LLM, creating a clear reasoning path. To refine this process, the framework employs an Alternate Self-reflection Training Strategy, which jointly optimizes the multimodal LLM and the question parser through reinforcement learning, enabling mutual refinement based on feedback. Evaluations on popular VideoQA datasets (NExT-QA, STAR, and NExT-GQA) show that MSR-ViR surpasses traditional and grounding-based methods, demonstrating improved accuracy and the ability to localize relevant video segments, thereby providing visually grounded evidence for its answers.

**Strengths:**

The paper presents several strengths in addressing Video Question Answering (VideoQA) tasks. Primarily, it enhances the interpretability of multimodal large language models (LLMs), which traditionally function as black-box systems. This framework, with its MoST-Grounding module, identifies relevant video segments and spatial regions, aligning them with text inputs to support answer derivation. The method also uses reinforcement learning to train the multimodal LLM and question parser LLM in tandem, represents an innovative approach that bolsters model transparency and clarity.

The paper summarized grounded and modular videoQA methods, which aim to answer questions while identifying relevant regions in the video, have seen notable advancements through the integration of multimodal large language models (LLMs).

The paper’s extensive experimental validation on datasets like NExT-QA and STAR showcases the method’s superior performance and its ability to provide visually-grounded evidence, setting it apart from existing models. This combination of interpretability, strategic training, and improved accuracy underscores the paper’s significant contributions to advancing VideoQA methodologies.

**Weaknesses:**

The use of reinforcement learning (RL) in the paper, while different in facilitating the Alternate Self-reflection Training Strategy, has some limitations concerning novelty and implementation. RL is well-established for optimizing policies in non-differentiable tasks, and its application to train LLMs collaboratively is not entirely unprecedented, as other multimodal and modular frameworks have explored similar strategies. Additionally, the reinforcement learning process depends heavily on the quality of intermediate feedback provided by the multimodal LLM, which may propagate errors if the initial predictions are suboptimal. Furthermore, RL’s computational overhead and potential convergence issues in complex scenarios are not fully addressed, leaving questions about its scalability for larger datasets or more intricate VideoQA tasks.

The paper claims the method is able to deal with complex tasks. Will you consider to add or discuss several recent video question answering tasks for comprehensive evaluations? (e.g., SOK-Bench: A Situated Video Reasoning Benchmark with Aligned Open-World Knowledge,  Complex-TV-QA: A Study of Situational Reasoning for Traffic Understanding, etc.)

The experimental improvements reported in the paper, while demonstrating the effectiveness of the proposed method, appear limited when compared to other state-of-the-art methods, as highlighted in Table 2. Although the MSR-ViR framework outperforms baseline and grounding-based approaches, the margin of improvement is relatively modest, raising concerns about the practical significance of the gains.

A concern regarding the paper is the absence of clear information about the availability of open-source code for review. It is crucial for verifying the implementation details, replicating experimental results, and evaluating the broader applicability of the proposed methods.

**Questions:**

NA

---

> ### Author Response · Authors · 2024-11-24
> **Responses to Reviewer gPWW**
>
> We sincerely thank the reviewer for taking time to review our paper and providing thoughtful feedback and insightful suggestions. We address the weaknesses and questions as follows:
>
> **Weakness1**
>
> Due to the lack of datasets with policy annotations, we are unable to directly perform supervised training for the question parser. As such, using feedback from the Multimodal LLM as rewards to train the question parser via reinforcement learning is a relatively straightforward approach. Based on the results of the ablation study (Table 3) and specific examples (Figures 4, 9, and 10), it can be observed that reinforcement learning enables the question parser to generate more reasonable policies that better support locating spatio-temporal segments relevant to the question in the video, demonstrating the effectiveness of our RL-based Alternate Self-reflection Training Strategy. We will experiment on larger VideoQA datasets with more complex scenarios in future work to evaluate the scalability of our RL-based training strategy.
>
> **Weakness2**
>
> We are sorry that, due to time constraints, we are unable to provide experimental results on the SOK-Bench and Complex-TV-QA datasets at this stage. Most questions in the Complex-TV-QA datasets focus on predicting future events based on the scenarios in the video, while the SOK-Bench includes many counterfactual reasoning questions. For these types of questions, grounding may not be helpful to improve Multimodal LLM to answer the question. As a result, they may not be suitable for evaluating our grounding-based framework MSR-ViR’s performance. It is worth noting that statistical analysis[1] shows that the average question length in the NExT-QA and STAR datasets is among the longest in commonly used VideoQA datasets, demonstrating the ability of our framework to handle long and complex questions effectively.
>
> **Weakness3**
>
> We test some more grounding-based model (SeViLa and GCG) and compare the results with our Llava-Next-based MSR-ViR framework, which are shown in the table below. As seen, MSR-ViR framework surpasses SeViLa and GCG on NExT-QA dataset and SeViLa on STAR-sub dataset. To further demonstrate the effectiveness of our framework, we test the base Multimodal LLM Llava-Next on NExT-QA and STAR-sub. As seen in the table below, the VideoQA accuracy has been significantly improved by utilizing MSR-ViR framework(from 73.1 to 74.9 on NExT-QA, from 69.9 to 71.0 on STAR-sub). What's more, for Acc@GQA and grounding metrics, MSR-ViR surpasses SeViLa on NExT-GQA dataset, showing a stronger grounded-qa capability.
>
> |                                 | NExT-QA Tem. | NExT-QA Cau. | NExT-QA Des. | NExT-QA Avg. | STAR-sub   Int. | STAR-sub Seq. | STAR-sub Avg. |
> | ------------------------------- | ------------ | ------------ | ------------ | ------------ | --------------- | ------------- | ------------- |
> | SeViLa                          | 69.4         | 74.2         | **81.3**     | 73.8         | 63.7            | 70.4          | 67.1          |
> | GCG                             | **72.6**     | 74.2         | 80.7         | 74.6         | -               | -             | -             |
> | Llava-Next                      | 69.5         | 73.3         | 79.7         | 73.1         | 67.6            | 72.1          | 69.9          |
> | MSR-ViR(ours, Llava-Next-based) | 72.2         | **74.6**     | 80.9         | **74.9**     | **68.9**        | **73.1**      | **71.0**      |
>
> |                                 | Acc@GQA  | mIoP     | IoP@0.3  | IoP@0.5  | mIoU     | IoU@0.3  | IoU@0.5  |
> | ------------------------------- | -------- | -------- | -------- | -------- | -------- | -------- | -------- |
> | SeViLa                          | 16.6     | 29.5     | 34.7     | 22.9     | 21.7     | 29.2     | 13.8     |
> | MSR-ViR(ours, Llava-Next-based) | **18.6** | **29.6** | **39.0** | **24.1** | **23.4** | **33.6** | **16.4** |
>
> **Weakness4**
>
> Our MSR-ViR framework is implemented based on the Swift[2] framework, and all experimental results are fully reproducible. We plan to open-source our code after the review and provide detailed documentation to facilitate the reproduction and application of our framework.
>
> [1] Fu, Chaoyou, et al. "Video-mme: The first-ever comprehensive evaluation benchmark of multi-modal llms in video analysis." arXiv preprint arXiv:2405.21075 (2024).
>
> [2] Zhao, Yuze, et al. "Swift: a scalable lightweight infrastructure for fine-tuning." *arXiv preprint arXiv:2408.05517* (2024).

---

> ### Author Response · Authors · 2024-11-27
> **A gentle reminder**
>
> Dear reviewer, thank you again for the review and we hope that our response and the uploaded revised paper have addressed your concerns. We would greatly appreciate your feedback and please feel free to let us know if you have any other questions.

---

> ### Author Response · Authors · 2024-12-01
> **A gentle reminder**
>
> Dear reviewer, thank you again for the review and the insightful feedback. We hope that our response and the revised paper have addressed your concerns. As the discussion stage is about to end, we would greatly appreciate your feedback and please feel free to let us know if you have any other questions.

---

### Author Response · Authors · 2024-11-26
**Explanation of the Paper Revisions**

Thank all the reviewers for taking the time to review our paper and providing valuable and constructive feedback. Based on the reviewers' suggestions, we have made several modifications and additions to the paper. The main revisions are as follows:

1. **Table 1**: We have added SeViLa and GCG as grounding-based baselines, along with the experimental results of our Llava-Next version of MSR-ViR.
2. **Table 2**: Results of SeViLa and our Llava-Next version of MSR-ViR have been added.
3. Based on the revision of Table 1 and Table 2, adjustments have been made to the experiments setup descriptions in Section 4.1, as well as the descriptions of the experimental results in Sections 4.2 and 4.3.
4. **Appendix**: A new Section A.6 has been added, which includes experimental results on model parameters and inference speed, as well as an ablation study on the selection of temporal grounding models.

---

### Meta-Review · Area_Chair_N8Yd · 2024-12-22

**Metareview:**

This paper proposes a modular system for video quesiton-answering (VideoQA). The motivation for a modular framework is that it can be both more intepretable (by providing a reasoning trace) and accurate than a single-black box model. The authors employ a parsing LLM to generate programs from the input question, separate temporal- and spatial-grounding modules, and an answering LLM as well. The authors use DPO to train the parsing LLM, as an alternative to few-shot prompting of the LLM as used in prior works such as MoreVQA.

Reviewers appreciated the motivation of the approach, and the reasoning traces that can be produced by the system. Concerns were that the proposed approaches substantially increases the computaitonal cost (it takes about 6x longer to answer a quesiton), and the accuracy improvements do not justify the computational cost of the approach. Reviewer gPWW also pointed out that the authors have not evaluated their approach on more complex Video QA datasets where reasoning traces could be more useful.

The AC agrees that the experimental evaluation of the paper is weak: the datasets used by the paper (Next-QA, Next-GQA and STAR) are all outdated and created before LLMs started being widely used. And there are also recent VideoQA datasets which contain substantially longer and complex videos (such as Video-MME (which the authors cited in the rebuttal) and LVBench) which would be more suitable as they are more challenging, require more reasoning and grounding, and are not saturated like Next-QA, as they were created with (M)LLMs in mind. And although the authors have mentioned in their paper and rebuttal that a focus of the paper is improving interpretability of VideoQA models, the experiments in the paper mostly focus on accuracy. The ablations in the paper (Table 3; top part) only consider accuracy too.

The final decision is therefore to reject this paper. Authors are encouraged to improve the experimental evaluation, and to resubmit a revised version of this paper to a subsequent conference.

**Additional Comments On Reviewer Discussion:**

Please see above. Reviewer concerns were that the proposed approaches substantially increases the computaitonal cost (it takes about 6x longer to answer a quesiton), and the accuracy improvements do not justify the computational cost of the approach. Reviewer gPWW also pointed out that the authors have not evaluated their approach on more complex Video QA datasets where reasoning traces could be more useful.

---

### Decision · Program_Chairs · 2025-01-22

Reject